# Enhancing the Learning Effectiveness of University of Science and Technology Students through Flipped Teaching in Chinese-Language Curriculum

**Cheng-Chih Huang, Yen-Ling Lin * and Ching-Yen Ho**

Center for General Education, National Taichung University of Science and Technology, Taichung 40401, Taiwan; jimhuang@nutc.edu.tw (C.-C.H.); pj0049@nutc.edu.tw (C.-Y.H.)
* Correspondence: yenling@nutc.edu.tw

**Abstract:** Chinese is one of the most important global languages, but at some universities of science and technology, Chinese-language learning is unfortunately not valued by students. According to the results of the Program for International Student Assessment (PISA), organized by the United Nations every three years, although Taiwanese students' performance in reading improved from 2015 to 2018, many problems remain in the teaching field that lead to a lack of student interest in Chinese-language learning. We attempted to use flipped teaching to intervene in the Chinese-language curriculum at the university of science and technology at which we teach. This methodological process and procedure was used to plan teaching materials, and 36 weeks of teaching with a Chinese-language curriculum were offered to freshman students in one academic year. We then evaluated teaching effectiveness through the pre- and post-tests of students' language proficiency, obtained feedback from students through school-wide teaching evaluation questions, reviewed the teaching effectiveness of the Chinese-language curriculum for the academic year, and achieved significant improvements in students' reading skills with regard to language learning.

**Keywords:** flipped teaching; Chinese reading; team-based learning; Chinese-language testing; teaching evaluation

## 1. Introduction

The Chinese-language curriculum for freshmen in Taiwan covers two main components of language proficiency, the ability to read and the ability to express oneself, both of which are basic training requirements for students to be able to learn on their own. The Ministry of Education has been promoting the "Basic Language and Multicultural Competency Cultivation Program" since 2015. The "School-Wide Chinese Reading and Writing Curriculum Innovation Sub-Project," with life education as the core principle, encourages colleges and universities to break away from the traditional teaching model of a compulsory Chinese-language curriculum for freshmen, have a team of teachers work together to develop teaching materials and innovative lesson plans, and lead students to read and write in depth and to develop the ability to read, comprehend, and reflect in a holistic manner. This reflects the early advocacy of the "flipped teaching" concept. The Ministry of Education is currently implementing the "Intensive Cultivation Program for Higher Education", which also sets the Chinese-language proficiency of college students as a basic requirement for teaching, so that colleges and universities will consider the enhancement of students' Chinese-language proficiency as an important task for higher education in Taiwan.

The current learning trend of Chinese-language education in vocational schools (high-school level) is that students generally follow the need for further education and focus on the interpretation of words and phrases recited for exams, basic knowledge of Chinese literature, and stereotypes of the author's life, resulting in a lack of overall understanding

and consideration of the material. As a result, students come to the university with a flawed learning mindset and inadequate basic training, which leads them to choose to either avoid or withdraw from the university's Chinese-language course, which requires abstract thinking skills. Moreover, teachers' efforts are met with student indifference, causing them to lose enthusiasm for teaching. In terms of the curriculum structure, the authors' Chinese-language curriculum for freshmen at a university of science and technology, for example, is taught in two 100-min classes per week, and this time constraint prevents teachers from fully exploiting their strengths to stimulate students' motivation and interest in learning. The school at which the authors teach is a university of science and technology in Central Taiwan. Even though the students' Chinese-language reading and expression skills are poorer than those of typical university students, their ability can be said to be above average. If teachers fail to use two classes per week to carefully structure and organize the curriculum, they will easily encourage a flawed learning mindset among students that "Chinese-language classes are not important"; if students do not develop reading habits, the course content will quickly become oversimplified and unengaging. Zhu believes that students graduating from vocational schools have always lacked opportunities to develop humanistic literacy and that their reading habits and reading comprehension skills lag behind those of ordinary high-school students, which is a worrying learning phenomenon because students' reading ability will be directly related to their self-learning ability and even their future success [1].

To address the above-mentioned issues related to the teaching of the Chinese language, this study describes the use of a flipped teaching model to intervene in the teaching of a Chinese-language curriculum for freshman students at a university of science and technology and enhance students' participation in learning through detailed curriculum planning and changes in design and teaching methods. We propose to use the pre- and post-test results of the "University and College Students' Language Literacy Test" in Taiwan as objective data to evaluate students' learning effectiveness. We also propose to use the results of the annual teaching evaluation of the curriculum at the teaching school, together with the records and observations of the teaching process, to examine the actual effectiveness of flipped teaching in the Chinese-language curriculum.

## 2. Literature Review

The authors argue that, for first-year university students, the ability to think and express oneself through language is an important basic ability to convey one's ideas and communicate with others, as well as the key to reading and comprehending texts, as part of their Chinese-language curriculum. As mentioned above, however, freshman students entering a university of science and technology usually do not have a good foundation of reading, thinking, or language expression, but they continue their poor learning attitudes from the past. Li suggests that Taiwan's 12-year national education allows students to learn without the same mandatory incentives as college entrance exams, making them vulnerable to low academic achievement [2]. Valas indicates that students' poor reading and comprehension skills can lead to "learned helplessness" because of frustration with their grades [3], and Jiang argues that such a negative reaction would inevitably reduce teachers' enthusiasm for teaching [4]. From 2014 to 2018, approximately 17% of Taiwan's higher vocational school students failed the Chinese Language Proficiency Test, achieving only a C grade. Taiwan's 12-year national education system allows students to enter high school without examination. However, given traditional Chinese culture, becoming a student in the technical vocational system is not the first choice for candidates but is considered a last resort; therefore, it is normal for students in higher vocational education to demonstrate poor Chinese language proficiency [5,6]. The students at a university of science and technology are chiefly higher vocational students, and their learning of the Chinese language follows from their learning at middle school and senior high school, which naturally leads to the frustration of both teachers and students due to poor language foundation, a passive learning attitude, utilitarianism, and avoidance.

Regarding the teaching of the Chinese language, Chen studied the obstacles that students encounter in the process of reading and how teachers should assist in their study [7]. Alvermann, Smith, and Readena discussed the application of prior knowledge in educational strategies [8]. The research of Meyer, Brandt, and Bluth focused on the exploration of the strategy used to teach reading to ninth-grade students. The strategy focuses on following the organizational structure of the text to determine the important content to remember. Paris discussed issues related to dyslexia and provided advice on how to solve them [9]. Liao developed a teaching plan for reading comprehension to improve the reading comprehension ability of middle school students. A total of five strategies including "grammatical structure", "browsing prediction", "self-inquiry monitoring", "understanding adjustment", and "enhancing memory" were provided, and a mode and direction of reading comprehension teaching were sought through teaching experiments [10]. The theory and methods of Chinese reading teaching presented by Zhou, Gan, and Zhang discussed the role, categories, and ability of reading, and they provided a variety of teaching methods and training methods for reading skills, which are very valuable for reference [11].

Regarding the aforementioned problems in the educational field, the concept of the "flipped classroom", developed by Bergmann and Sams, may offer a solution. The flipped classroom is a "learner-centered" educational concept; from its inception to its promotion, the meaning of the "flipped classroom" pedagogy has been expanding, and almost any teaching model that reverses the tradition, is hybrid, or is subject-neutral falls into the category of the "flipped classroom" pedagogy, which is now prevalent in various subjects in elementary schools, middle schools, high schools, and even universities in Taiwan. Bergmann and Sams attempted to combine online instruction and offline learning to enhance students' motivation and the effectiveness of supplementary learning when missing classes [12]. Chen also proposed the idea of "changing the teacher-driven learning in the traditional classroom to group discussion and listening of students as well as guidance and conclusion of the teacher" [13]. With the prevalence of this teaching method, teachers have tried to use various teaching materials and methods to enhance students' self-learning ability, and flexibility in students' learning experiences and testing methods has increased. Scholars have identified four main steps of "flipped teaching" in practice: students need a pre-study mechanism; teachers ask questions and guide them in class; students complete assignments in class; and teachers review work immediately, design activities, and lead students to deeper learning. This framework provides a reference for pedagogues in Taiwan [14]. Butt suggests that the core concept of the flipped classroom is the idea of moving the instructional material outside of formal class time and using formal class time to engage students in collaborative and interactive activities related to this material. Thus, the authors would survey students at both the beginning and end of the semester to obtain their views on the structure of general teaching and the flipped classroom. The results of this study showed that, after they experienced the entire course in this way, students' views on the flipped classroom approach became more positive [15]. O'Flaherty and Phillips suggest that the flipped curriculum has the ability to build lifelong skills for 21st-century learners. Abeysekera and Dawson propose a flipped classroom approach that eliminates traditional didactic instruction and replaces it with active classroom tasks and before-/after-school assignments. This study attempted to construct a theoretical argument that the flipped approach may increase student motivation and help them to manage their cognitive load, but researchers are called upon to conduct more specific research on the effectiveness of the flipped classroom approach [16,17]. Zuber affirmed the need for further research on flipped classrooms and supported the development of an educational methodology [18].

McLean et al. evaluated students' adjustment to flipped classrooms and concluded that flipped classrooms potentially lead to greater educational gains. Following the implementation of flipped classrooms, students developed independent learning strategies, spent more time on tasks, and engaged in profound and active learning [19]. Akçayır

and Akçayır conducted a systematic review of the literature on flipped classrooms and summarized the advantages and challenges in flipping classroom instruction [20].

In the field of Chinese-language teaching in Taiwan, the "share-start teaching method" proposed by Cheng, a high-school teacher, based on the concept of flipped classrooms, has recently emerged as a new teaching model for teachers of the Chinese language at elementary and middle schools [21]. It has been promoted in Taiwan for many years, and several academic research papers support its teaching effectiveness. It has been revealed that the share-start teaching method has had a significant impact on the Chinese-language reading comprehension skills of high-school students, and action research on the teaching of the Chinese language to high-school students also achieved remarkable teaching effectiveness. In case studies of Chinese-language teachers' implementation of the share-start teaching method, the method was proven to be effective in improving students' oral expression, writing skills, and the atmosphere of classroom learning. Most studies were limited to the effectiveness of Chinese-language teaching at elementary, middle, and higher vocational schools, while neither the application nor a review of this method for teaching the Chinese language at universities is yet available. Therefore, the present study examines the effectiveness of flipped classroom teaching on the reading and writing skills of freshman students at a university of science and technology [22–24].

## 3. Methodology

### 3.1. Teaching Methods

To solve various problems in teaching first-year college students the Chinese language, this study adopted the concept of flipped teaching to design and implement the curriculum. The study is based on four main steps of flipped teaching: students' pre-study, teachers' in-class questions and guidance, students' in-class assignments, and teachers' immediate review. Moreover, this study adheres to the principle of flipped teaching, and the course is conducted in such a way that the class is not a solo performance by the teacher but leaves approximately 30–50 min for students to discuss and think in small groups, ensuring that the students have the opportunity complete the pre-class reading and discussion, thus enhancing the learning effect in class.

### 3.2. Research Methods

This study used the results of the pre- and post-tests of the "University and College Students' Language Literacy Test" to examine teaching effectiveness, and the overall teaching curriculum design was integrated to implement the flipped teaching strategy. In the two-credit freshman Chinese-language course, students were allowed to develop their reading, analysis, integration, writing, and communication skills through group reading, group discussion, presentation, reporting on reading highlights, extended thinking, and reading and essay presentation in midterm/final exams. During the teaching process, feedback sheets and essays on learning experiences were used to collect students' learning experiences and feelings about the teaching content so that the teaching could be adjusted in a timely manner. Eventually, the results of "Curriculum and Instruction Statistics" at the university at which the authors taught were corroborated with the results of the aforementioned language proficiency test. We use two measurements in this study. The first is a teaching opinion survey: the school at which the authors serve conducts a survey on students' satisfaction with teachers' teaching. The questions are designed by the school and are divided into two categories: closed questions and open questions. The survey also permits open answers to questions about whether teachers' teaching attitudes, teaching content, teaching methods, and evaluations are fair. It has been implemented for over 20 years and is a teaching feedback survey conducted by colleges and universities across Taiwan at the end of each semester. All schools adopt anonymous answers, and most of the questions are the same. It has a certain level of public credibility, and all teachers adopt the survey results when demonstrating their own teaching effectiveness.

The second instrument is the "University and College Students' Language Literacy Test". This literacy test is mainly based on a reading test and writing test. It is the "National Platform Program for Chinese Language Literacy Test", established and implemented by the National Taichung University of Education in Taiwan in 2013. The purpose is to share it with universities and colleges across Taiwan through the establishment of Chinese language literacy test resources and standards. Built with standardized evaluation tools, this test can establish a fair and objective norm of ability and place the students participating in the test on a measurement scale of national university and college students' Chinese language literacy ability, so that the test scores can have reference value. As this Chinese language literacy test for reading and writing has been established in Taiwan for seven years, the test platform provides examination questions and rating, so teachers cannot predict what the test questions are and how to rate them. As it allows pre-test and post-test comparisons at the same level, it is of reference value for teachers' teaching effectiveness.

### 3.3. Research Subjects

Two classes of freshman students from the College of Commerce at the authors' teaching school, where the authors teach, received 36 weeks of Chinese-language teaching in two sessions per week for two semesters, where the flipped teaching pedagogy was applied. The subjects had participated in the unified entrance test for the four-year and two-year programs in Taiwan's science and technology colleges and had a certain level of achievement. Classes were divided in a random S-shape, so the two classes of research subjects had a certain degree of impartiality and objectivity each year. In this study, we adopted the original classroom implementation of the pedagogical intervention model and did not reassign classes. The same teaching materials, methods, and strategies were provided to the two classes participating in the project, and the focus of the observation was on whether the flipped teaching method could improve the reading skills of the two classes of freshman Chinese-language students.

### 3.4. Curriculum Framework and Implementation Steps

#### 3.4.1. Curriculum Framework

The training of language skills needs to be conducted over time. The freshman Chinese-language course planned in this study comprised four units (Development of Basic Self-Learning Skills and Patterns, Communication and Expression, A Better Self, and Choice of Values), each with 20 selected texts, and the training period was 36 weeks for the first and second semesters of the first year of college. Beginning with a model for building students' self-learning skills, students practiced mastering the meaning of words, developing a sense of the language, and thinking through simple and interesting texts. This section was designed to remedy their basic language skills and build the students' confidence by allowing them to overcome their fears of classical Chinese texts through group interpretation of the texts, as well as to guide their thinking and their search for answers through questioning. Through reading gains in the text and note-taking, students were able to distinguish between "rational cognition" and "perceptual cognition". Rational cognition requires distinguishing information, comparing, summarizing, judging, analyzing, inferring, and integrating before having the opportunity to apply. In contrast, perceptual cognition conveys the construction of an ideal world through the depiction of objects in a work and allows students to express their subjective experience and self-consciousness through their thoughts, emotions, images, insights, and meditations elicited by the text. Through repetitive exercises, students will establish a basic pattern of self-learning and extend it to each subsequent unit, deepening and broadening their reading of the text.

#### 3.4.2. Teaching Steps

Since there were only two 50-min classes per week, the authors' model of flipped teaching for the first-year college Chinese-language course was to assign reading texts and

use group work to allow students to complete the readings in advance of class; in addition, discussion questions were provided for initial extended thinking within the group. In the formal classroom, the group reported on their understanding and extended thinking on the group reading (completing the process of learning, thinking, discussing, and expressing), while the teacher provided explanations of the text and integrated the groups' perspectives and final conclusions. At the end of the course, the students were given questions to reflect on and asked to express their ideas in "Five Minutes of Handwriting". The operation of group reading and classroom discussion is shown in Table 1.

**Table 1.** Operation of group reading and classroom discussion.

| Teaching Steps | Conducting Activities | Expected Training/Results Expected Training/Results |
|---|---|---|
| Group learning | Group reading and discussion of questions before class. | • Through group reading, students' patience and interest in participation are enhanced, and their willingness to continue reading is strengthened.<br>• Reading is conducted in groups to reduce misreading and increase comprehension.<br>• The students will also be able to appreciate the different perspectives of their peers during the reading process.<br>• Grouping is heterogeneous, and each group has a good leader to lead the group in reading and completing tasks before class.<br>• A KPI (group discussion sheet) is designed for peer-to-peer grading. |
| Group reporting | Each group should complete the above discussion and the group report within the time limit each time they read the text. | • Train students to be patient and polite by listening to other groups' reports.<br>• Design a KPI mechanism for grading and announce each group's strengths and weaknesses, scores, and suggestions in each unit to establish healthy competition among groups and motivation to participate in reading.<br>• The group report should be accompanied by a simple PowerPoint presentation. Students will be trained to integrate key points and oral expression skills. |
| Lecturing | Provide explanations of prior knowledge for the text in progress, integrate the views of each group, guide their extended thinking on the unit, and offer an overall conclusion after reading the text. | • Design classroom learning notebooks, plan the key points of the notes, and regularly check the contents of the notebooks to develop students' ability to listen to the lectures and grasp the key points for note-taking.<br>• The questions presented in each group report are used to adjust/correct students' extended thinking or misunderstandings.<br>• Connect the common themes between the units to guide students to think more deeply.<br>• Depending on the topic, in-class discussions or group activities will be opened up to sustain students' attention. |
| Five-minute impromptu writing mechanism | After completing the reading and extended thinking, the topics are handwritten, so as to reinforce the impression of learning without being bound by the content. | • Each text or topic is completed, and five minutes are set aside to guide students to write and train them to express themselves smoothly and accurately.<br>• For each impromptu writing assignment, the teacher gives positive feedback to form a text discussion mechanism between the teacher and each student and forge a sense of identity with subject learning. |

**Table 1.** *Cont.*

| | | |
|---|---|---|
| Evaluation of outcomes | Midterm/final exams for each semesterPre- and post-tests for language proficiency testing Administration of teaching feedback questionnaires at the end of the semester in the Office of Academic Affairs (anonymous) | • The midterm/final exams of each semester are mainly based on reading ability and writing ability, and the learning effectiveness is checked and evaluated on a regular basis.<br>• Language proficiency tests are administered once each semester (before the first semester of teaching and after the second semester of teaching). The test is administered outside the Ministry of Education, and the results are more credible.<br>• The feedback questionnaire at the end of the semester in the Office of Academic Affairs allows students to express their feelings about learning. |

The above-mentioned curriculum is in line with the basic concept of flipped teaching, and some changes have also been made according to the needs of the curriculum. These changes are intended to break away from the traditional teaching mode whereby the teacher is the sole leader and discussant and to provide more space for students' participation and self-learning. The basic framework for curriculum design is to "make up for the students' deficiencies"—when students do not have enough to read, they should increase the amount of reading; when the written expression is not enough, their opportunities to practice should be increased, but the overall framework should not deviate from a consistent goal.

3.4.3. Teachers' Observations and Students' Unit Teaching Feedback

(1) Themes of course units

Taking the unit of "Literature and Emotion" in the authors' curriculum as an example, the authors followed the themes of the three emotions most commonly encountered in people's lives, namely family affection between family members, romantic love between lovers, and friendship between friends. The authors selected one representative short literary article as the main reading text for each theme and introduced several extended readings to provide each group with choices.

(2) Interactive operations between the teacher and students

The teacher first brought out the theme with a video, introduced the author and the content of the work, and led the extended thinking on the topic, which lasted around 30–35 min. Then, in each class, there were eight groups of students who introduced the text that they read together in their group, the problems that they identified in the text, and the conclusions of their discussion. Each group's talk lasted for around 3–5 min, and altogether they took around 35–40 min to complete. Then, the teacher summarized and classified the problems found by each group and related them to the theme. For example, family problems most easily reflected complaints such as generational differences and poor parent–child communication. Through the description of the text and the students' reflections, the authors hoped that students could see the importance of family affection and where the conflicts lie and would be encouraged to think calmly about how to deal with conflicts, so as to minimize them and make their parents feel that their children were growing up. This part took around 20 min, and the final 5 min were reserved. Guided by the pre-designed questions, students could gather their thoughts and write about their feelings.

(3) Students' feedback on course units

The authors designed a feedback sheet for each unit to collect students' written feedback, and 10 feedback opinions on the above teaching units were randomly selected from the 2 classes:

1. It's great. I learned about a lot of books that I haven't read.
2. I have a deeper understanding of friendship, family affection, and romantic love.

3. In this unit, no matter whether family affection, romantic love, or friendship, every emotion made me think a lot and learn a lot.
4. I didn't like it when I learned that I had to report in groups after the first class, because I didn't know the students in the group at all, but I got used to the feeling of grouping, and even felt that I had learned a lot.
5. I learned the attitudes involved in friendship, romantic love, and family affection.
6. This topic is sensitive to me, so I have more feelings about it.
7. These stories are very helpful to living and life. We can make our own choices based on the reflection in the course.
8. This unit gave me a deeper understanding that literature is full of all kinds of emotion.
9. It provided a rare chance for me to seriously examine my emotional situations from the past to the present, as well as the method and mentality of interactions with people.
10. I found that the affections I have, whether it is family affection, romantic love, or friendship, don't come easy.

### 3.5. Research Hypothesis

This study is a quasi-experimental design, and the proposition to be examined is as follows:

"The application of the flipped teaching method to first-year college students' Chinese-language instruction is directly related to the improvement of students' Chinese reading comprehension." On the basis of this hypothesis, the pre- and post-test scores of the research subjects participating in the Ministry of Education's University and College Students' Language Literacy Test in Taiwan were compared to each other, and the post-test results showed improvements, which is the key to examining the validity of the hypothesis.

## 4. Research Results

### 4.1. Pre-Test/Post-Test Effectiveness of Language Proficiency

During the 2020 academic year, two classes of students were the research subjects, and approximately 44–47 students participated in each class in the pre- and post-tests of the University and College Students' Language Literacy Test.

4.1.1. Improvements in Reading Proficiency in the Pre- and Post-Tests of the University and College Students' Language Literacy Test in Taiwan

Figure 1 shows that 47 students participated in the pre-test in the first-year Class 1 of the Department of Accounting Information during the 2020 academic year, with the largest number of students in the low and middle range of scores (below 80). After post-test results were observed, the number of students in the low-score range decreased to two; the number of students scoring 81–90 increased to 29; and the number of students in the high-score range increased to nine. It can be seen that the overall reading ability of the students significantly improved.

The results in Figure 2 show that 44 students participated in the pre-test in the first-year Class 2 of the Department of Accounting Information in the 2020 academic year, with nine students scoring less than 70, 16 students scoring less than 80, and 17 students scoring less than 90. Only two students achieved high scores in the range of 91–100. After the post-test results were observed, the number of students in the low-score range decreased to two; the number of students in the 81–90 score range increased to 21; and the number of students in the nearly perfect score range increased to 11. Overall, students' reading ability can be judged to have improved significantly.

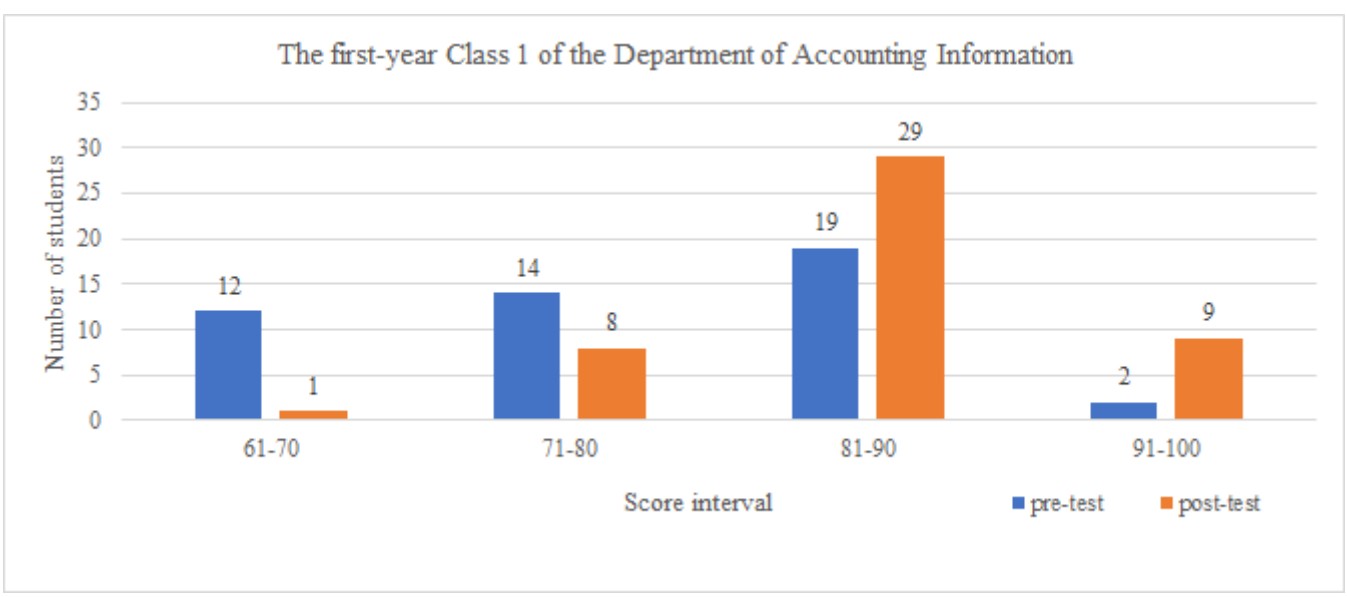

**Figure 1.** Pre- and post-test performance of the first-year Class 1 of the Department of Accounting Information in the 2020 academic year.

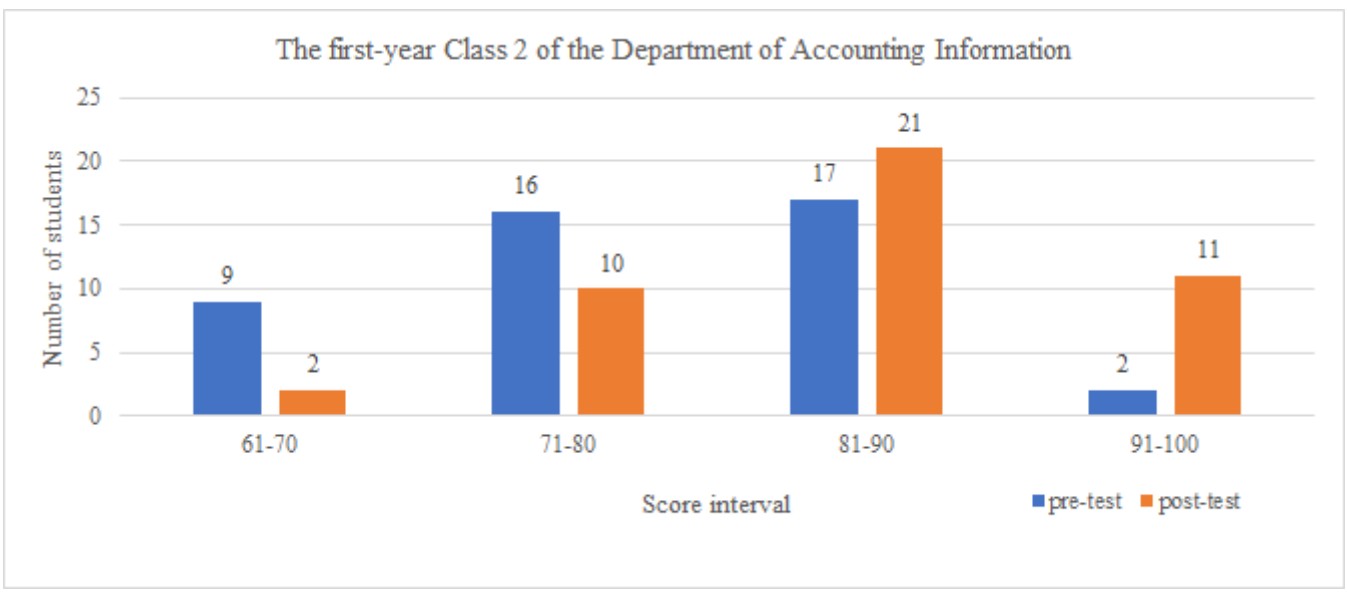

**Figure 2.** Pre- and post-test performance of the first-year Class 2 of the Department of Accounting Information in the 2020 academic year (2020).

The above pre- and post-test data from the University and College Students' Language Literacy Test show that the quantitative performance of the "flipped teaching" significantly influenced the teaching of freshman Chinese language at the authors' teaching school.

4.1.2. Performance of the Accounting Information Department in the Pre-Test and Post-Test of the Reading Examination

As can be seen from the reading evaluation scores of language literacy from 2019 to 2020 presented in Table 2, the reading evaluation scores of Class 1 and Class 2 in the Accounting Information Department were average. Before the implementation of flipped teaching, the improvement rate of Class 1 was 0.61%, while that of Class 2 was −0.50%. The improvement rate was not large, and there was even a drop. One year into the implementation of flipped teaching, the improvement rate of Class 1 was 9.39% and that of Class 2 was 9.33%, with significant improvements in both classes. From the pre-test

and post-test data in the above table, it can be seen that the effect of flipped teaching on the success of Chinese-language teaching activities for first-year students at the authors' teaching school was significant. The above-mentioned pre-test and post-test data were provided and collated with the authorization of the Chinese Language Literacy Testing Center of National Taichung University of Education.

**Table 2.** Performance of the Accounting Information Department in the pre-test and post-test of the reading examination.

| Language Literacy Reading Evaluation Score in 2019 | Pre-Test | Post-Test | Improvement Rate |
|---|---|---|---|
| Average score of Class 1 in the Accounting Information Department | 88.72 | 89.26 | 0.61% |
| Average score of Class 2 in the Accounting Information Department | 88.72 | 88.28 | −0.50% |
| **Language Literacy Reading Evaluation Score in 2020** | **Pre-Test** | **Post-Test** | **Improvement Rate** |
| Average score of Class 1 in the Accounting Information Department | 76.68 | 83.88 | 9.39% |
| Average score of Class 2 in the Accounting Information Department | 76.92 | 84.10 | 9.33% |

Data source: Compiled by the researchers based on data from the Chinese Language Literacy Testing Center of National Taichung University of Education.

### 4.2. Teaching Evaluation of School Curriculum

4.2.1. Content of the Teaching Evaluation Questionnaire

To understand teaching effectiveness, the school where the authors teach administered the "Teaching Opinion Survey" (Appendix A) for each subject to each student in each semester.

The questions were answered anonymously and were designed to survey students' opinions on the courses that they were studying and to provide teachers with reference material for improving their teaching. Each question was scored on a 5-point scale, with 5 points for "strongly agree", 4 points for "agree", 3 points for "moderate", 2 points for "disagree", and 1 point for "strongly disagree". Teachers were evaluated in terms of four major dimensions: teaching attitude, teaching methods, teaching content, and assignments and evaluation, for a total of 12 items. There was also space for open-ended comments, which students could complete within a limit of 100 words.

4.2.2. Evaluation of the Teaching of the Chinese Language in the 2020 Academic Year

The teaching attitude of teachers was assessed for the 2020 academic year (Table 3) using a 5-point scale, and the scores for the two classes were 4.34 and 4.58, respectively. Nearly 90% of the students agreed that the teachers were dedicated, serious, and respectful in terms of teaching attitudes. The majority of students accepted the flipped teaching model and believed that the teaching could stimulate their interest in learning and fulfil the intended teaching objectives. The teaching content was systematic, pragmatic, and unique, and met the future needs of the students; the learning assessment mode of the subjects was diverse and fair.

In the open-ended opinion survey, Class 1 respondents commented, "Although I felt annoyed by many previous reports, I really feel that I have increased my knowledge of the Chinese language, and the teachers are really dedicated and serious about the lessons" and "I think it is not bad to let students preview the curriculum through flipped teaching." Class 2 respondents said, "The way the teacher conducts the class is great, and we can learn a lot."

The above survey echoed the results of the pre- and post-tests of the language proficiency test in the 2020 academic year, which indicated that most students were positive about the flipped concept of teaching freshman-level Chinese.

**Table 3.** Reference data for Chinese-language teaching evaluation in the 2020 academic year.

| Reference Data for Chinese-Language Teaching Evaluation in the 2020 Academic Year | | |
|---|---|---|
| I. Teaching attitudes | Class 1 | Class 2 |
| 1. The teacher fulfilled the required teaching hours (18 weeks per semester) and was neither late nor left early. | 4.43 | 4.71 |
| 2. The teacher was dedicated and responsible and actively guided students. | 4.39 | 4.51 |
| 3. The teacher respected the individual differences of students (such as ability, ethnic group, gender, etc.) | 4.2 | 4.51 |
| II. Teaching methods | | |
| 4. The teacher clearly conveyed the content of the lesson. | 4.26 | 4.53 |
| 5. The teacher used teaching media (such as slides, multimedia, teaching aids, reference materials, etc.) to increase their interest in learning. | 4.28 | 4.47 |
| 6. The teacher guided students to collect information, think independently, solve problems, and express opinions. | 4.39 | 4.56 |
| III. Teaching content | | |
| 7. The teaching content was arranged systematically and had learning value. | 4.13 | 4.4 |
| 8. The teaching content was pragmatic and distinctive and met the needs of students. | 4.07 | 4.27 |
| 9. The teacher followed the course outline and completed the scheduled progress | 4.22 | 4.47 |
| IV. Assignments and evaluation | | |
| 10. The teacher carefully corrected and reviewed the students' homework and examination papers. | 4.37 | 4.64 |
| 11. The teacher used multiple assessment methods (such as homework, exams, reports, work presentations, evaluation of learning attitudes, etc.) | 4.24 | 4.53 |
| 12. The teacher's assessment method was fair and reasonable | 4.11 | 4.42 |
| Total average | 4.26 | 4.5 |

## 5. Limitations and Discussion

Although the teaching results show that the use of the flipped teaching method was effective in the field of freshman Chinese-language teaching at the authors' teaching school, some shortcomings and limitations remain in the implementation of the flipped teaching method, requiring improvements and adjustments. The following questions and suggestions will shape future improvements in the implementation of this teaching method and serve as a reference for those who are interested in reversing the Chinese-language curriculum at the university level in the technical college system.

(1) During the actual implementation of the method, we noted the importance among teachers of establishing a learning consensus with students at the beginning of the semester. If the explanation is inadequate, teachers and students are prone to friction resulting from maladjustment and poor communication, and it is not easy to establish a good learning environment.

(2) In the team-based learning session, teachers can treat group leaders as important helpers in teaching and give them the utmost importance and trust. During the teaching period, teachers should hold regular appointments (after each midterm/final exam) and design "group discussion sheets" to develop a sense of commitment and

responsibility among group leaders. This section will allow for quicker resolution of many individual students' learning difficulties.

(3)     When designing the curriculum and teaching materials, teachers should think in terms of "students' needs" and should not teach only what they want to teach. Teaching materials that can be connected to students' life experiences are easily accepted. It is advised to open up space to allow students to give feedback on their individual experiences and ideas to form a positive cycle of teaching and learning, in order to achieve a more harmonious teacher–student relationship and smooth teaching.

(4)     The "five-minute impromptu writing" section requires considerable patience. The author insists on handwriting, hoping that the connection between the hand and the brain will help students to capture their ideas quietly and express them accurately. This creates a heavy workload because the teacher's feedback needs to be sent back to the students immediately (usually every week), and between the writing and the feedback, the teacher and the students form a textual dialogue that brings them closer together and sustains their willingness to learn.

(5)     The peer review form is an important mechanism in training students to listen quietly to their classmates' reports because they are required to give their opinions on each group's report on stage. All groups had to concentrate on listening to the lecture, so they reduced their use of their mobile phones. However, this mechanism requires weekly evaluation and suggestions for each group to be used as the basis for the next improvement of the group, and it also requires the teacher's patience.

(6)     Good learning experiences and results involve a whole year of patient waiting and persuasion. The stereotype of freshman-level Chinese in students' minds cannot be easily erased or changed in the classroom with a few words, but the effectiveness of language learning is difficult to see immediately in the short term, and the results are not easily verified. As a result, patiently accompanying students and constant explanations become a daily reality in class. Even if students feel that the teacher is extremely verbose, they have to be humble and persevering in completing the planned course curriculum.

(7)     The major aim of Chinese language teaching is to improve the reading and expression of the language. Through the test, the results can be seen. Chinese-language teaching also has teaching purposes, such as cultivation of reading habits, self-study ability, and moral character. These can be accumulated gradually through the design of teaching processes and the determination of a class mode. Therefore, the authors repeatedly expressed the reading mode of the text in the thesis, i.e., guiding reading through asking questions; training students to listen, think, and express themselves through the mechanism of five-minute casual writing; note-writing in peer review form; and a group discussion. If the participants generally perform better in the post-test than in the pre-test, it indicates that the intervention via the teaching method was effective.

## 6. Conclusions

Looking at the teaching effectiveness of an academic year, we can confirm that the authors' intended goal of flipping teaching was achieved. By allowing students to participate in the pre- and post-tests of the University and College Students' Language Literacy Test, teachers can obtain objective feedback on the curriculum design and teaching methods, and students can also reflect on their reading ability and learning outcomes from the results of the pre- and post-tests. Through the statistical presentation of the "Teaching Opinion Survey" conducted by the school, the authors' teaching was recognized by the majority of students. The statistical data of the survey concurred with the quantitative results of the pre- and post-tests of the University and College Students' Language Literacy Test conducted by the Ministry of Education.

The team-based learning mechanism in the course design allows students to read, discuss, and present their understanding and insights on stage every week, as well as to

perform the "five-minute impromptu writing" exercise. Each training session can effectively accumulate and improve students' reading and expression skills, with appropriate "extended thinking" of the text and questions that require close reading and deep thinking to answer. This allows students to connect the content of the text with abstract issues such as their own moral and existential values, leading them to develop independent thinking skills and to demand more of themselves, as well as to enhance teamwork and interpersonal harmony.

The grouping mechanism gave rise to peer care and friendship in the classroom. Before each week's class, the group members were concerned about their classmates' attendance in the same group, and over time, the rate of skipping class was reduced; students' participation in class was improved; and the frequency of mobile phone use was reduced, which was an unexpected gain that encouraged students to be more attentive and engaged in class.

Through weekly text readings (including extended readings) and problem discussions, students will be able to develop their self-learning skills through multiple exercises and will be interested in finding answers to their questions. In addition, helping students to establish good study habits and experiences will enhance students' identification with the school and give them the opportunity to reposition the value of the subject "Freshman Chinese language" in their minds.

The core of the flipped teaching strategy is that teachers are not limited to one-way teaching; instead, the strategy treats the students as the protagonists of learning in the classroom, giving them at least 30 to 50 min of each lesson. Teachers act as guides and moderators and try their best to complete a series of learning, discussion, and expression processes in the classroom.

The foregoing data can validate the following conclusions:

(1) The use of the flipped teaching model enhances students' interest in learning and improves their attitudes in class;
(2) The flipped teaching model provides space for students to participate in discussions and reflections, increasing their learning effectiveness;
(3) The pre- and post-test reading scores of the Ministry of Education's University and College Students' Language Literacy Test showed improvement in students' reading ability;
(4) The survey of students' opinions on teaching showed that students were highly receptive to the flipped teaching strategy, and their opinions were quite positive.

**Author Contributions:** All authors made significant contributions throughout this piece of research and agreed to submit the manuscript in the current form. Y.-L.L. made major contributions in terms of writing, designing questionnaires and analyzing data. C.-C.H. and C.-Y.H. contributed in terms of conceptualization and revising the manuscript. All authors have read and agreed to the published version of the manuscript.

**Funding:** This research received no external funding.

**Institutional Review Board Statement:** Not applicable.

**Informed Consent Statement:** Informed consent was obtained from all subjects involved in the study. Written informed consent has been obtained from the patient(s) to publish this paper if applicable.

**Data Availability Statement:** Not applicable.

**Conflicts of Interest:** The authors declare no conflict of interest.

**Appendix A**

**Table A1.** Teaching opinion survey.

| I. Teaching attitudes |
| --- |
| 1. The teacher fulfilled the required teaching hours (18 weeks per semester) and was neither late nor left early.<br>2. The teacher was dedicated and responsible and actively guided students.<br>3. The teacher respected the individual differences of students (such as ability, ethnic group, gender, etc.) |
| II. Teaching methods |
| 4. The teacher clearly conveyed the content of the lesson.<br>5. The teacher used teaching media (such as slides, multimedia, teaching aids, reference materials, etc.) to increase their interest in learning.<br>6. The teacher guided students to collect information, think independently, solve problems, and express opinions. |
| III. Teaching content |
| 7. The teaching content was arranged systematically and had learning value.<br>8. The teaching content was pragmatic, distinctive, and met the needs of students.<br>9. The teacher followed the course outline and completed the scheduled progress. |
| IV. Assignments and evaluation |
| 10. The teacher carefully corrected and reviewed the students' homework and examination papers.<br>11. The teacher used multiple assessment methods (such as homework, exams, reports, work presentations, learning attitudes, etc.)<br>12. The teacher's assessment method was fair and reasonable |
| V. Other comment |
| Please fill in within 100 words |

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
