# Peer review of "Enhancing the Learning Effectiveness of University of Science and Technology Students through Flipped Teaching in Chinese-Language Curriculum"

_sustainability, doi:10.3390/su13179743_

Round 1
Reviewer 1 Report
Thanks for presenting this paper on Chinese language teaching. It's interesting to apply flipped teaching into Chinese language curriculum. However, I have some questions and concerns regarding the paper.
- I would suggest to add a section with brief introduction of the measurement you used in the study. For example, the teaching opinion survey, is it self developed by the author or had been previously applied in other studies? Is it a valid and reliable instrument? Also please provide a description on the national college students language literacy test.
- I noticed that the study was took place in 2020, so could you please specify whether the class was administered online due to Covid? If so, what might be some potential impacts on the implementation of the curriculum? Or it's still a regular face to face class.
- The control group seems missing in the study. The current study only implements pre-post design. It is not strong enough to make the conclusion that the teaching effectiveness is achieved with the experimental approach and curriculum.
- What are students' feedback toward the flipped teaching method? Both students and teachers are important components of the classroom, it would be better if the study could provide students' behavior or feedback in the class. For example, use classroom observation to capture students' academic behavior and teacher dosage of instruction under the new method.
Reviewer 2 Report
The use of technology in teaching Chinese literacy is a welcomed topic but I am afraid that the study has serious limitations as a quasi-experimental study. I am not entirely sure if it can be called a quasiexperiemental study. I suggest that the authors attend the following issues:
- Did the study have a control group? I am afraid that I could not find the relevant information. The pre and post-test results cannot confirm whether the pedagogical experiments worked or not. I understand that it may not be ethically permissible, feasible for the authors to have a control group, but the way they present the study has to be changed. It is not about the impact of pedagogical intervention on the participants' Chinese literacy learning. It is more about how the participants' Chinese literacy develops in a context of this pedagogical experiment and how/why the development happened.
- The statistical analysis (pre and post test results) is very basic. I am afraid that not much significant claims can be based on the analysis. It is not clear whether these changes have sufficient statistical significance (how about effect sizes)?
- I also think that the authors should focus on whehter the evaluation results help explain the developments in Chinese literacy as documented by the two tests. The authors should help readers to work out what elements of the pedagogical efforts might have caused the participants to do something to improve their literacy skills.
- The authors really need to tell readers in what sense the findings constitute significant knowledge contributions to the field before they talk about limitations and potential implications that the findings have.
Round 2
Reviewer 1 Report
Thanks for your revision.
Author Response
Thank you for giving us the opportunity to strengthen our manuscript with your valuable comments and queries. We have worked hard to incorporate your feedback and hope that these revisions persuade you to accept our submission.
Sincerely,
Chingyen, Ho
Reviewer 2 Report
I feel unsure of this claim about experimental intervention without a control group but I also feel that the authors have a point here.
I suggest that the institutional details in the abstract and manuscript be removed to anonymize the study.
I also feel that the literature the authors engaged with is a bit insufficient. It is necessary for them to review more recent relevant studies on teaching Chinese language and literacy
The connection to the journal's title (sustainality) needs to be established explicitly in the manuscript, too.
Author Response
Thank you for giving us the opportunity to strengthen our manuscript with your valuable comments and queries. We have worked hard to incorporate your feedback and hope that these revisions persuade you to accept our submission.Please see the attachment.
Sincerely,
Chingyen, Ho
